# A Novel Algorithm to Detect White Flowering Honey Trees in Mixed Forest Ecosystems Using UAV-Based RGB Imaging

Atanas Z. Atanasov [1,*], Boris I. Evstatiev [2], Valentin N. Vladut [3] and Sorin-Stefan Biris [4,*]

1 Department of Agricultural Machinery, Agrarian and Industrial Faculty, University of Ruse "Angel Kanchev", 7017 Ruse, Bulgaria
2 Department of Electronics, Faculty of Electrical Engineering, Electronics and Automation, University of Ruse "Angel Kanchev", 7017 Ruse, Bulgaria; bevstatiev@uni-ruse.bg
3 National Research—Development Institute for Machines and Installations Designed to Agriculture and Food Industry, 013813 Bucharest, Romania; vladut@inma.ro
4 Faculty Biotechnical Systems Engineering, National University of Science and Technology POLITEHNICA Bucharest, 006042 Bucharest, Romania
* Correspondence: aatanasov@uni-ruse.bg (A.Z.A.); sorin.biris@upb.ro (S.-S.B.)

**Abstract:** Determining the productive potential of flowering vegetation is crucial in obtaining bee products. The application of a remote sensing approach of terrestrial objects can provide accurate information for the preparation of maps of the potential bee pasture in a given region. The study is aimed at the creation of a novel algorithm to identify and distinguish white flowering honey plants, such as black locust (*Robinia pseudo-acacia*) and to determine the areas occupied by this forest species in mixed forest ecosystems using UAV-based RGB imaging. In our study, to determine the plant cover of black locust in mixed forest ecosystems we used a DJI (Da-Jiang Innovations, Shenzhen, China) Phantom 4 Multispectral drone with 6 multispectral cameras with 1600 × 1300 image resolution. The monitoring was conducted in the May 2023 growing season in the village of Yuper, Northeast Bulgaria. The geographical location of the experimental region is 43°32′4.02″ N and 25°45′14.10″ E at an altitude of 223 m. The UAV was used to make RGB and multispectral images of the investigated forest massifs, which were thereafter analyzed with the software product QGIS 3.0. The spectral images of the observed plants were evaluated using the newly created criteria for distinguishing white from non-white colors. The results obtained for the scanned area showed that approximately 14–15% of the area is categorized as white-flowered trees, and the remaining 86–85%—as non-white-flowered. The comparison of the developed algorithm with the Enhanced Bloom Index (*EBI*) approach and with supervised Support Vector Machine (SVM) classification showed that the suggested criterion is easy to understand for users with little technical experience, very accurate in identifying white blooming trees, and reduces the number of false positives and false negatives. The proposed approach of detecting and mapping the areas occupied by white flowering honey plants, such as black locust (*Robinia pseudo-acacia*) in mixed forest ecosystems is of great importance for beekeepers in determining the productive potential of the region and choosing a place for an apiary.

**Keywords:** UAV; remote sensing; classification; clustering; apicultural plants

## 1. Introduction

The application of unmanned aerial vehicles (UAVs) for remote monitoring of the growth of crops has become a widely adopted practice in modern agriculture. The use of UAVs in agriculture allows farmers to gather a large amount of information about vegetation indices [1–4], content of nitrogen (N), phosphorus (P), and potassium (K) in plant fodder [5], soil salinity [6] contamination with hazardous toxic elements in soil, and vegetation [7], soil water deficit [8,9], plant water status [10], etc.

Apart from agriculture, UAV is successfully used in forest surveys for the purpose of effective management of forest resources. The study of individual forest species in

broad-leaved forests is essential in order to estimate forest parameters. According to [11,12], Individual Tree Detection (ITD) is a crucial initial process in conducting a comprehensive forest inventory, encompassing tree position identification, species classification, crown delineation, biomass estimation, and the prediction of various attributes related to individual trees. The detection of individual trees in deciduous forests [13] presents a new individual approach with graph analysis based on UAV-derived point cloud data. Some shortcomings related to the reliability of the data obtained from the UAV when monitoring newly reforested low-growing saplings have been avoided by the "Divide-and-conquer" (DAC) and the local maximum (LM) method used in [14]. The information obtained from remote forest monitoring enables the creation of detailed maps for the distribution and classification of individual forest species in specific regions. An interesting study was conducted by [15] who used data from a RedEdge-MX sensor to obtain multiple types of features and multiseasonal images and help identify key driving factors that are conducive to tree species recognition and mapping. In another research [16], a technique is introduced that utilizes a custom-built toolbox within ArcGIS, powered by ArcPy, to autonomously detect trees in high-resolution data captured from UAVs. This toolbox, named TreeDetection, incorporates a specific tool called TreeDetect, which necessitates three input parameters: a raster file, a conversion factor, and an output folder. In [17], a method for the estimation of pistachio Tree Canopy Volume using planar area and ground shadows calculated from UAV RGB images was presented. An interesting study was conducted in [18] describing the use of a single UAV platform with three different imaging sensors for three common applications in precision viticulture: missing plant detection using an RGB camera, vegetative vigor monitoring using a multispectral camera, and water stress detection using a thermal camera.

The large-scale capabilities of remote sensing can also be used in beekeeping, offering vegetation maps that highlight the occurrence and flowering periods of plants highly appealing to bees [19]. The presence of sufficient area and species diversity of flowering vegetation in a given area is crucial for the sustainable development of bee colonies [20]. To obtain high yields of honey and for the pollination of agricultural crops, it is important to calculate the nutritional capacity of a given area within the flight distance of bee colonies [21]. Detecting and determining the areas with flowering agricultural crops, such as sunflower, canola, lavender, and others, is often a routine task related to obtaining information about the sown areas directly from farmers or information obtained from global monitoring systems. Unlike agricultural crops, distinguishing flowering plants from non-flowering plants and determining their areas in mixed forests and grasslands is a difficult task requiring very precise high-resolution spatial imagery. Such images can be collected by UAVs equipped with high-resolution sensors [22]. Different machine learning algorithms are mentioned in the literature for detecting and recognizing apicultural plants, such as Random Forest (RF), Classification and Regression Trees (CART), Support Vector Machine (SVM), and others [23–25].

Interesting observations about detecting and distinguishing apicultural plants *Thymus capitatus* and *Sarcopoterium spinosum*, using UAV, were made in [26], whose training method distinguished the two plants with 70% accuracy. The optimal flight parameters and processing options for the detection of taxus and olive trees with UAV-Borne Imager are presented in [27]. In [28], an approach to assess the color coverage in grasslands using UAV RGB imagery was shown. This study proves the concept that flower cover is an important aspect of habitat quality for bees. The systematic monitoring of flowering forest species and flowering grasses is of great importance for beekeeping in determining the optimal placement locations for beehives. Furthermore, of great importance for the sustainable development of bee colonies is the competition between them in areas overpopulated with apiaries. Along with monitoring flowering plants, UAVs can be used to detect apiaries in certain overpopulated areas. The data obtained from them are the basis for the development of mathematical models for optimal placement of bee colonies [29]. The geographical location of the individual regions, the altitude, and the prevailing agrometeorological conditions

predetermine the rich variety of flowering forest and meadow vegetation in the various beekeeping regions.

An important factor in beekeeping is the recognition of the blooming status of apicultural plants, for which different approaches exist, one of which is based on the application of vegetation indices (VI). They are based on different spectrum imaging data, such as RGB, red, near-infrared, etc., and allow the creation of classification maps that highlight numerous factors, such as chlorophyll content, vegetation density, canopy structure, etc. [30–32]. In [33], it was proposed that the Enhanced bloom index (*EBI*) for the identification of areas with blooming vegetation based on RGB images can be determined using:

$$EBI = \frac{R + G + B}{\frac{G}{B}.(R - B + 256)} \tag{1}$$

It was compared with two other approaches: SVM classification and the simple Brightness index (*BI*). The obtained results showed that *EBI* was able to separate the bloom areas from soil and other vegetation efficiently.

In [34], a method for the identification and monitoring of blooming Mikania micrantha using UAV-obtained RGB images. Six vegetation indices were tested (Over-green index—EGI; Normalized over-green index—NEGI; Blue-green differential index—BGDI; Greed-red differential index—GRDI; Normalized green-red differential index—NGRDI; and Plant pigment ratio index—PPR) yet none of them performed well. Better results were achieved with a multi-scale segmentation approach, where PPR and Brightness were used as parameters for a deep learning algorithm that achieved 88.62% pixel recognition accuracy.

In [35], a method to identify the flowering of orchard trees based on the Brightness index was used:

$$Brigthness = \frac{R + G + B}{3} \tag{2}$$

The methodology relies on white, gray, and black reference panels installed in the investigated area and compares the panels' brightness with the trees' brightness. The selected flowering threshold is pixel brightness equal to or greater than the average panel brightness. A similar approach was used in [36], where white, black, and 6 grayscale panels were used for calibration. The study used UAV-obtained multispectral images (RGB, red edge, near-infrared) to identify trees and their crowns. Additional layers were generated based on Gaussian blur, Normalized difference vegetation index (NDVI), Normalized difference red edge index (NDRE), and Brightness and near-infrared (NIR) Brightness. Thereafter, algorithm-based classification maps were created, allowing the refinement of the tree crown extent, to identify the tree crown centers and to map the individual tree crowns. The achieved overall accuracy surpassed 96%.

Another approach is based on different machine learning algorithms, which are used for object-based identification. In [37], a deep learning approach was used to monitor the flowering of different tropical tree species. The ResNet-50 deep learning algorithm was used with UAV-obtained RGB images. The flower classification pixels were obtained with a recall of 85.8% and a precision of 95.3%. In [38], UAV-obtained RGB images were used to estimate the blossom count and density in an orchard. The authors used the MeshLab 2020.07 software tool, where two criteria were selected—the white blossom color threshold and the blossom grid filter size. Their optimal values were estimated using 420 regressions, the coefficient of determination, and the root mean square error. Flower mapping was also used in [39]; however, in this case, an object-based approach with the R-CNN neural network was applied. For training, testing, and validation regions of interest were annotated using polygons, keeping only the pixels of interest. The precision and recall for some of the flowers was higher than 90%, while for others the algorithm did not perform well due to lack of enough training data. An object-based approach was also used in [40], where You Only Look Once (YOLO) v4 was applied to identify the flowering rate of litchi. The analysis is based on UAV-obtained RGB images of the litchi orchard,

which were used to train the CNN for different resolutions of the input image. Thereafter, a polynomial model was made to correlate the number of identified flower clusters with their actual number.

Beekeeping is one of the rapidly developing industries in Bulgaria. The largest number of bee colonies are located in the hilly plain in the central and north-eastern part of the country, where large areas of rapeseed (*Brassica napus*) and sunflower (*Helianthus annuus*) and diverse forest flowering vegetation, such as dogwood (*Cornus mas*), hazel (*Corylus*), black locust (*Robinia pseudo-acacia*), hawthorn (*Crataegus*), and linden (*Tilia cordata)* prevail. One of the most honeyed forest plants is the black locust (*Robinia pseudo-acacia*) [41]. The production of nectar is highly influenced by the weather conditions in the area. In certain years, the flowers might produce minimal to no nectar. Even in favorable nectar-producing conditions, the time when the flowers are in bloom is brief, lasting from 7 to 14 days. This emphasizes the need for bee colonies to be well-prepared to make the most of this short window. The black locust (*Robinia pseudoacacia*) is among the frequently encountered invasive tree species in Europe [42]. In Bulgaria, it covers roughly 21.2% of the entire forest land [43]. In some areas of the country, the black locust (*Robinia pseudoacacia*) forms uniform forest stands, while in others, individual trees are found scattered within forests dominated by broadleaf species such as oak (*Quercus*) and linden (*Tilia cordata)*; therefore, the discovery of the black locust among the other forest species is of significant importance for beekeeping. The climatic diversity of the local regions and the environmental diversity require local experimental research.

Given these characteristics, monitoring techniques and training methodologies should be tailored to each specific situation. This highlights the need for creating appropriate honey tree detection algorithms for different regions.

The primary goal of this study is to create a novel algorithm to identify and distinguish white flowering honey plants such as black locust (*Robinia pseudo-acacia*) and to determine the areas occupied by this forest species in mixed forest ecosystems using UAV-based RGB imaging.

## 2. Materials and Methods

### 2.1. Study Area

A study was conducted in the region of Razgrad near the vicinity of Yuper village, north-eastern Bulgaria (Figure 1) in the 2023 growing season. The coordinates of the village are 43°54′28.59″ N, 26°23′49.02″ E, and it stands at an elevation of 107 m above sea level. The predominant forest vegetation is composed of broad-leaved mixed forests. Important for beekeeping are the shrubs dogwood and hawthorn and the tree species of black locust and linden. The climate is continental with characteristic cold winters and hot summers. The soils in the area are leached chernozem. The most important honey plant in the area is the black locust. A large part of black locust trees are found as single trees in mixed forest massifs of oak and linden, which makes it difficult to determine the areas they occupy when calculating the honey balance in the beekeeping region. Based on phenological observations made in the period from 2021 to 2023, we found that the beginning of flowering in 2021 started on 15 May 2021 and lasted until 22 May 2021. In 2022, flowering started on 15 May 2022 and ended on 25 May 2022. In 2023, flowering started on the 18 May 2023 and ended on the 30 May 2023. Multiple aerial monitoring procedures were carried out, and for the purpose of our experiment, photographs were collected during the period of full bloom of the black locust coinciding with the date 21 May 2023.

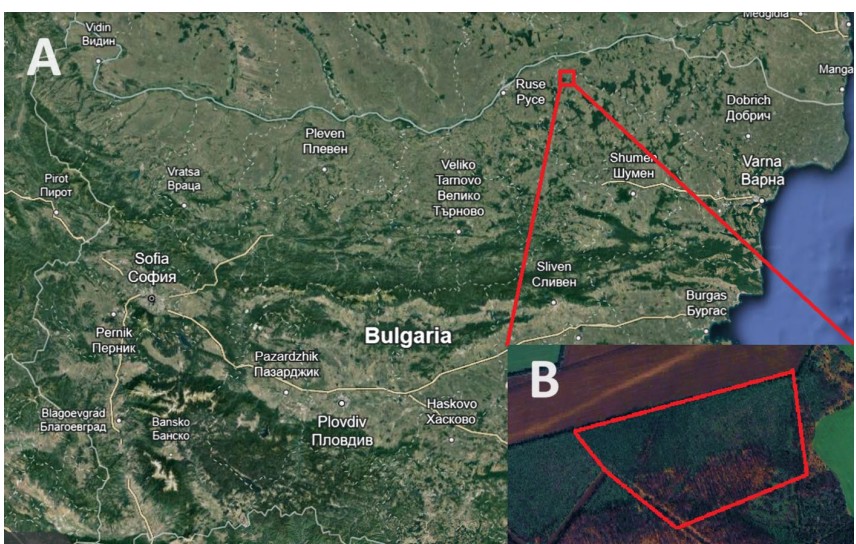

**Figure 1.** Location of the experimental plot: (**A**) geographic location of the experimental area in the Northeastern Bulgaria, Razgrad region; (**B**) experimental forest south of the village of Yuper.

*2.2. UAV Images Acquisition*

A P4 multispectral drone (Figure 2), developed by DJI (Da-Jiang Innovations, Shenzhen, China) was employed for the tasks of capturing images of blooming black locust trees situated in diverse forest regions near the village of Yuper. P4 multispectral is a quadcopter with vertical take-off and landing capabilities, able to hover at low altitudes. The maximum flying speed is 50 km h$^{-1}$ in P-mode and 58 km h$^{-1}$ in A-mode. The drone's maximum flight time is approximately 27 min.

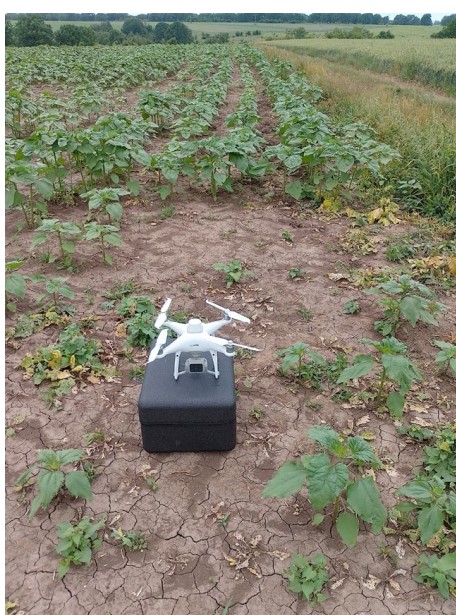

**Figure 2.** The DJI Phantom multispectral drone.

The drone is equipped with six 1/2.9″ Complementary Metal Oxide Semiconductor (CMOS) sensors, including one RGB sensor for capturing visible light images and five monochromatic sensors for multispectral imaging. Each sensor is effective at 2.08 MPixels (total of 2.12 MPixels). The filters for the five monochromatic sensors for multispectral imaging are as follows: Blue (B): 450 nm $\pm$ 16 nm; Green (G): 560 nm $\pm$ 16 nm; Red (R): 650 nm $\pm$ 16 nm; Red edge (RE): 730 nm $\pm$ 16 nm; Near-infrared (NIR): 840 nm $\pm$ 26 nm.

The image resolution is 1600 × 1300 px (4:3.25), the Controllable tilt ranges from −90° to +30°, and the ground sample distance (GDS) is H/18.9 cm/pixel. It supports two image formats: JPEG (visible light images) + TIFF (multispectral images) [44].

The flight routes were planned using the DJI Pilot (version v2.5.1.17) software, and operations were carried out from 12:00 a.m. to 13:00 p.m., in order to minimize shading. The other parameters of the flight are: 64 m relative altitude, 90° course angle, 70% front overlap ratio, and 60% side overlap ratio. Considering the GDS characteristic of the drone, the approximate dimensions of the observed area should be less than or equal to 54.2 m × 44.0 m, because the tree crowns are elevated above the ground surface.

The weather conditions during the flight were determined with the assistance of a portable Meteobot® agrometeorological station (Meteobot, Varna, Bulgaria), installed in the village of Yuper. During filming, the average wind speed measured 1.67 m s$^{-1}$, the average atmospheric humidity stood at 51.23%, and the average solar radiation was 513 W m$^{-2}$. The P4 multispectral camera settings were configured to capture images at even intervals aligned with the aircraft's path of travel. A total of 2117 images were taken using a multispectral UAV, producing three orthophotos.

*2.3. Algorithm Development*

Before an algorithm is developed, the following requirements are defined for it:

✓ **It should be able to identify white blooming trees**—the primary goal of the study is to identify white flowering honey plants; therefore, this requirement directly aligns with the core objectives. To achieve this, the algorithm must be capable of recognizing the characteristic features associated with white blooming trees, such as the color of their flowers;

✓ **It should filter out small blooming areas**—small blooming areas may represent partially blooming trees, or they could be other bright objects unrelated to the target honey plants (e.g., withered trees, small rocks). Filtering out these small areas is important to ensure high accuracy of the results. This way it will help to eliminate noises and focus the analysis on the significant blooming clusters, preventing potential misinterpretation of the data;

✓ **It should create clusters, representing the different blooming areas**—identifying individual blooming areas and creating clusters is essential for a more detailed and nuanced analysis of the distribution of white blooming trees. Clustering allows for the differentiation of distinct patches of blooming vegetation within the mixed forest. This requirement enhances the ability to understand the spatial distribution and arrangement of the target honey plants;

✓ **It should estimate the total area of these clusters**—the ultimate goal is not only to identify individual blooming areas but also to quantify the extent of the white blooming trees' distribution in the mixed forest. Estimating the total area covered by the identified clusters provides valuable information about the abundance and spatial coverage of the honey plants, contributing to a comprehensive understanding of their presence in the ecosystem.

Considering the object of the investigation is wild-growing black locust trees, the object-based approach for the identification of blooming trees is not appropriate. The main reason is that the black locust blossom is too small to be identified as a separate object and there are too many of them overlapping each other. Therefore, we decided to use a color-based approach, which is similar to the VI one. A general overview of the proposed algorithm is presented in Figure 3. It can use visual spectrum (RGB) data from a UAV or a satellite source. Next, the images are processed and their pixels are classified into two categories:

1. White bloom (WB) pixels—they represent a blooming part of the trees;
2. Not white bloom (NWB) pixels—they represent everything else.

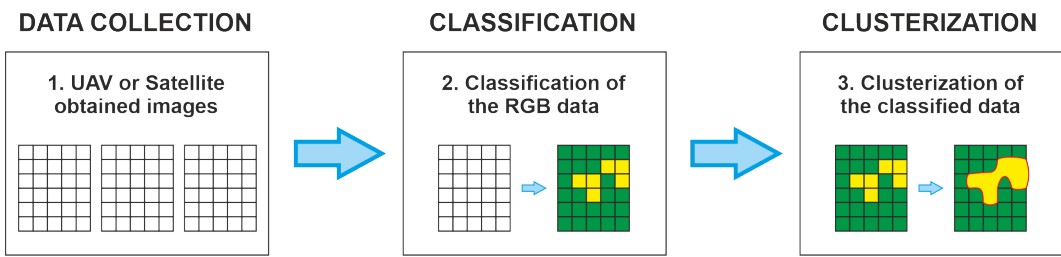

**Figure 3.** Overview of the proposed algorithm.

In the next stage, the created classification map is clustered, in order to obtain the zones with the blooming trees.

A more detailed description of the proposed algorithm is presented below. It includes the following steps:

**Step 1.** The pixels are classified in the WB and NWB categories, using the following criteria:

$$px_{WB} = (R > 170) \text{ and } (G > 200) \text{ and } (B > 150) \tag{3}$$

$$px_{NWB} = (R \leq 170) \text{ or } (G \leq 200) \text{ or } (B \leq 150) \tag{4}$$

where *R*, *G*, and *B* are the red, green, and blue components of each RGB pixel and take values from 0 to 255. The threshold values have been chosen experimentally and are aimed at selecting bright pixels with green components because the blooming black locust has leaves.

**Step 2.** WB pixel groups, where the number of pixels is less than a certain threshold $N_{thr}$ are filtered out; i.e., classified as NWB (Figure 4).

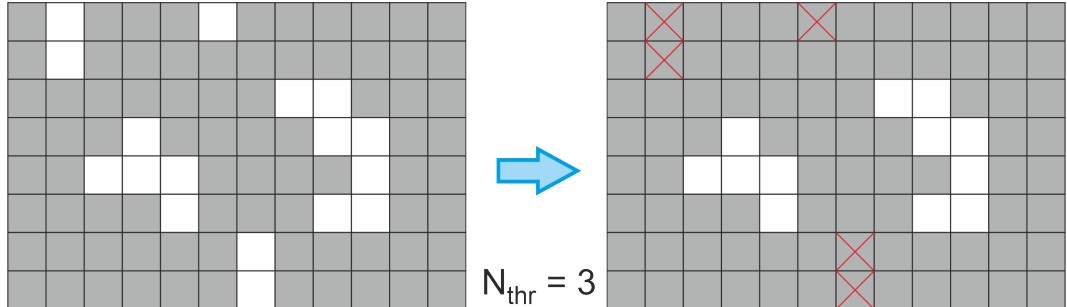

**Figure 4.** Example for filtering out WB pixel groups with threshold $N_{thr} = 3$.

**Step 3.** The WB pixels are clustered by adding a circle around the center of each pixel with radius $R_B$ (Figure 5). All intersecting circles are combined together into a single cluster.

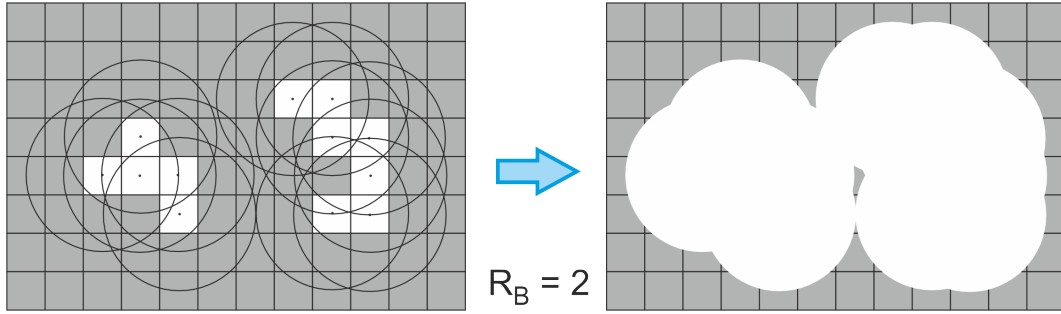

**Figure 5.** Example for clustering the WB pixel with radius $R_B = 2$.

**Step 4.** Any NWB areas, which are located inside a WB cluster are filtered out, as shown in Figure 6.

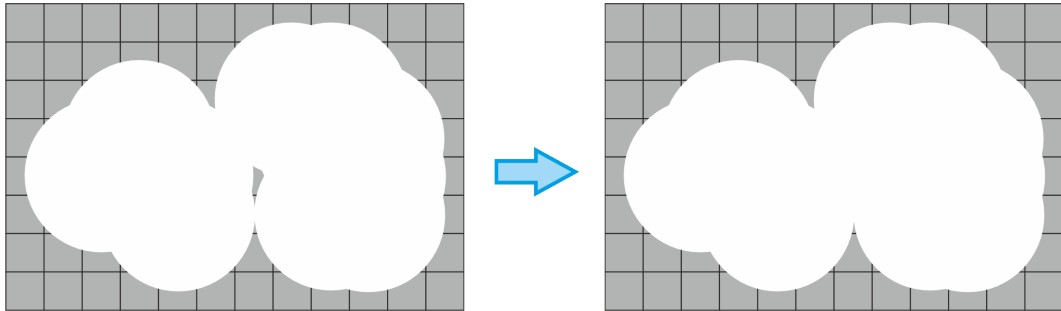

**Figure 6.** Example for filtering out NWB areas, which are surrounded by a WB cluster.

**Step 5.** The final step of the algorithm is to estimate the percentile area of the WB clusters, compared to the whole area of interest.

The QGIS v3.32 software, which is a project of the Open Source Geospatial Foundation (OSGeo, Oregon, United States), has been chosen for the implementation of the developed detection algorithm. The main reasons for its selection are that the tool is free, open source, and platform independent; furthermore, it provides a wide range of data processing instruments, which can be used in a user-friendly graphical environment. The following steps have been used to implement the developed methodology:

**Step 1.** An image is loaded in the software and the classification of the pixels is carried out using the "Raster calculator"—if the image is called "img.jpg", then the raster calculator, i.e., Equation (3), is implemented as follows:

$$("img@1" > 170) * ("img@2" > 200) * ("img@3" > 150) \tag{5}$$

**Step 2.** In order to filter out WB groups with less than $N_{thr} = 3$ pixels, the Sieve tool is used with "Threshold" 3.

**Step 3.** Next, the created classified raster is converted to a polygon, using the "Contour polygon" tool with default settings.

**Step 4.** In order to implement the clustering with radius $R_B = 10$ pixels, the "Buffer" tool is used with Distance = 10 and default remaining settings.

**Step 5.** Next, the NWB gaps within the WB clusters are filtered out using the "Delete holes" tool, with the "Remove holes with area less than" property set to 0 (i.e., all holes should be removed).

**Step 6.** Finally, in order to obtain the percentage of WB clustered area, the "Field calculator" is used with the following equation:

$$area(\$geometry) * 100/sum(area(\$geometry)) \tag{6}$$

The values of $N_{thr}$ and $R_B$ have been selected experimentally and are appropriate for the current flying height of the UAV (64 m). If the height and/or the camera resolution change or if satellite images are used, other values might be more appropriate.

## 3. Results and Discussion

### 3.1. Data Processing

Two of the drone images have been selected (Figures 7a and 8a), in order to test the developed algorithm and compare it with other approaches. They were selected so that different situations from the environment are included, such as withered trees, differently colored trees, a field near the forest, etc. The two images were processed using the raster calculator (Step 1 of the implementation), and the results from the preformed classification are presented in Figures 7b and 8b, respectively. It can be seen that the identified white bloom pixels are quite scattered and therefore cannot be directly used for analyzing the images. Therefore, the next steps from the image processing have been applied.

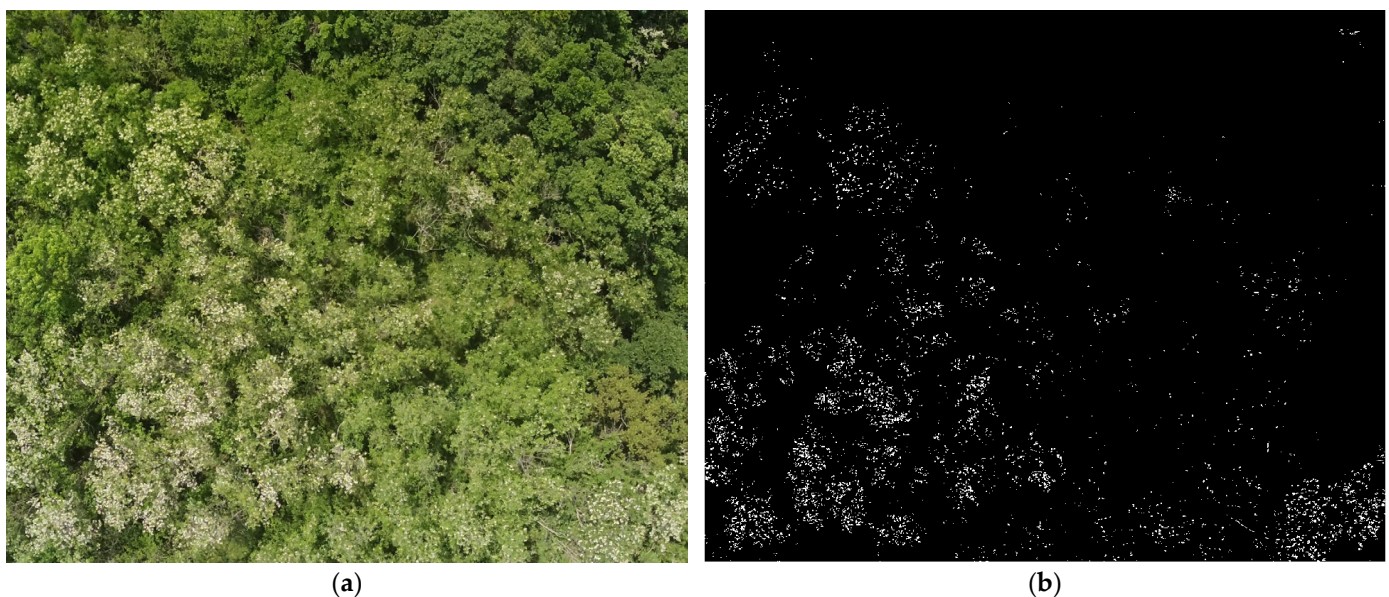

**Figure 7.** The first testing image (**a**) and the results from the performed classification of white blooming black locust trees (**b**).

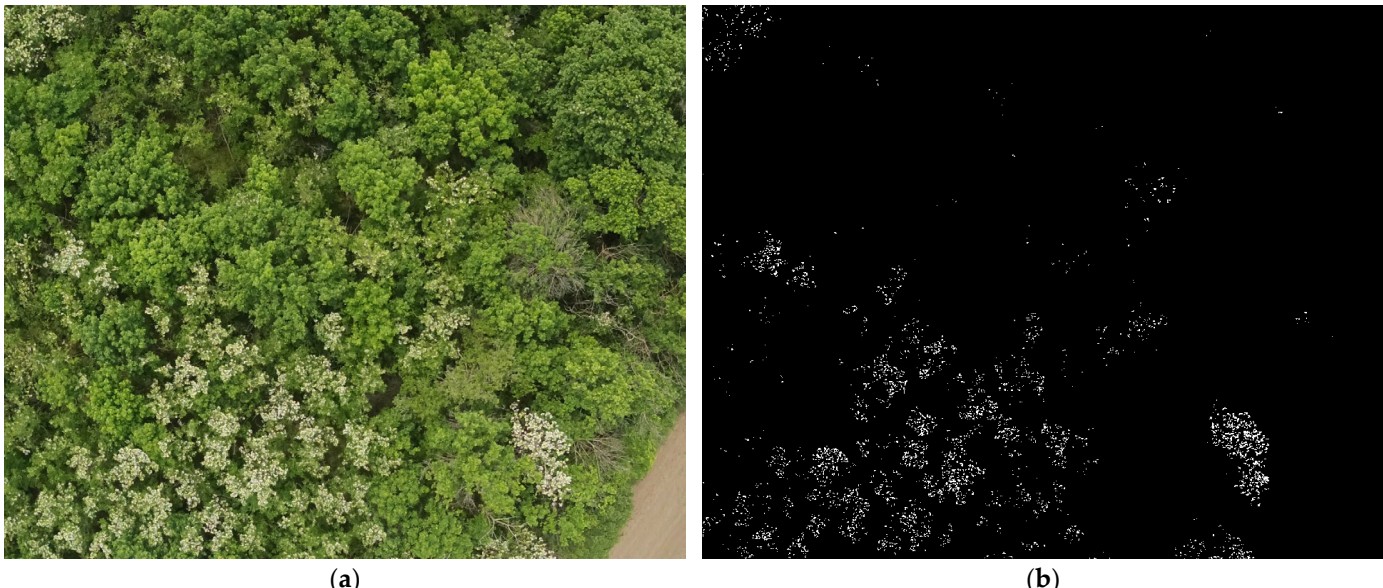

**Figure 8.** The second testing image (**a**) and the results from the performed classification of white blooming black locust trees (**b**).

Figure 9 presents the results from step 5 of the implemented algorithm, where classified images have been clustered. Each cluster is marked with a red line, surrounding an area that is identified as WB. For convenience, the clusters are shown on top of the original images, which allows easy evaluation of the algorithm performance.

Next, the accuracy of the performed classifications has been assessed. A total of 82 regions of interest (ROI) were selected from the first testing image and 101 from the second. The main criterion when selecting the ROIs was that they are undisputable; i.e., areas that cannot be clearly identified as either blooming or non-blooming were not used. Thereafter, 10,000 points were randomly chosen within the selected ROIs using the stratified approach—the number of points within each class is proportional to its relative area. The obtained confusion matrices for the two testing images are presented in Tables 1 and 2. It

can be seen that the obtained Cohen's Kappa coefficients are 0.88 and 0.95, respectively for the first and second images.

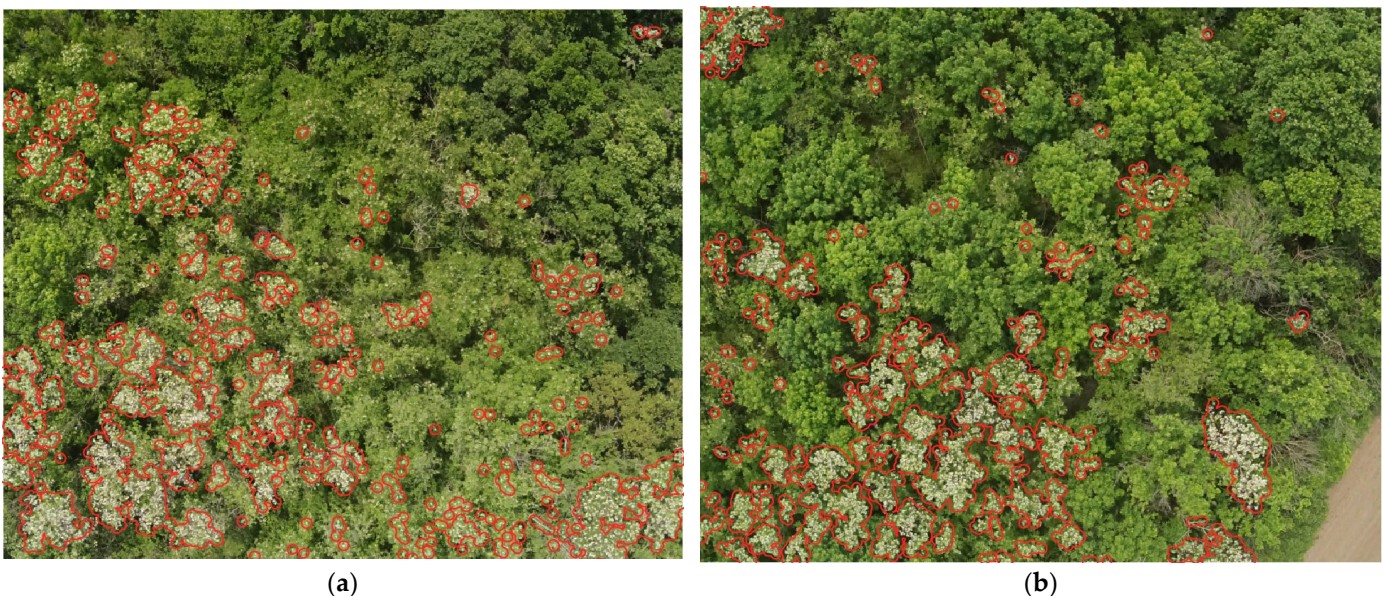

(**a**)            (**b**)

**Figure 9.** Results from the clustering of the classified images with the proposed criteria and algorithm for the first (**a**) and second (**b**) testing images.

**Table 1.** Confusion matrix for the classification of the first testing image using the developed algorithm.

| | | Reference | | |
|---|---|---|---|---|
| | | Not a White Bloom | White Bloom | Precision |
| **Result** | **Not a white bloom** | 7897 | 227 | 0.972 |
| | **White bloom** | 144 | 1732 | 0.923 |
| | **Recall** | 0.982 | 0.884 | **Kappa: 0.88** |

**Table 2.** Confusion matrix for the classification of the second testing image using the developed algorithm.

| | | Reference | | |
|---|---|---|---|---|
| | | Not a White Bloom | White Bloom | Precision |
| **Result** | **Not a white bloom** | 8303 | 134 | 0.984 |
| | **White bloom** | 9 | 1552 | 0.994 |
| | **Recall** | 0.999 | 0.921 | **Kappa: 0.95** |

Several important observations could be made:

1.  The developed algorithm for the identification of white blooming trees operates fairly well and is able to identify the blooming black locust tree clusters efficiently. The Kappa values (0.88 and 0.95), obtained for the two images, indicate almost perfect agreement.
2.  The precision and recall of the "not a white bloom" class do not fall below 0.972, which indicates a very high rate of successful classification.
3.  The recall of the "white bloom" class is slightly lower (0.884), which indicates that there were some false positives when identifying the "not a white bloom" class.
4.  Some areas of the photos that contain soil or withered trees are not identified as white blooming trees, which corresponds with our requirements.
5.  Small blooming areas are filtered out, which corresponds to the defined requirements.

6.  Some of the black locust trees are only partially beginning to bloom and are not identified as blooming yet.

Finally, the last stage of the implemented algorithm is performed (Step 6), where the percentile area of each class is evaluated. The obtained results are summarized in Table 3. It can be seen that approximately 14–15% of the area is categorized as white blooming trees and the remaining 86–85%—as non-white bloom.

**Table 3.** Percentile results from the classification of the two images.

| Experimental Object | White Bloom, % | Not a White Bloom, % |
|---|---|---|
| Image 1 | 14.55 | 85.45 |
| Image 2 | 13.79 | 86.21 |

*3.2. Comparison of the Results with Other Approaches*

Considering the object-based classification is not directly applicable to wild black locust forests, there are two possible approaches that could be used for comparison: appropriate vegetation indices and pixel-based classifications. In order to be able to directly compare the accuracy of the developed algorithm with the other two approaches, the same testing images were used.

1.  *Comparison with the Enhanced Bloom Index*

To the best of our knowledge, the *EBI* (Equation (1)) is the only index, capable of highlighting white blooming trees, and therefore it has been selected as a comparison method in our study. According to [33], values of *EBI* 0.61 ± 0.10 correspond well with the white color of flowers in trees; however, in our study, these values were not able to properly identify the blooming area and returned numerous false positives. It could be speculated that the difference comes from the lack of green leaves in the original study while blooming black locusts have leaves. Therefore, we have chosen another threshold for the classification of the pixels as either blooming or non-blooming. If *EBI* > 1.5 then the pixel is considered to be blooming otherwise non-blooming. Finally, the presented algorithm to be used with the QGIS v.3 software has been implemented by replacing the Step 1 equation with:

$$((``img@1'' + ``img@2'' + ``img@3'') * (``img@3'') / (``img@2'' * (``img@1'' - ``img@3'' + 256)) > 1.5) \quad (7)$$

The obtained clusters for the two images are presented in Figure 10 and the confusion matrices from the two classifications—in Tables 4 and 5.

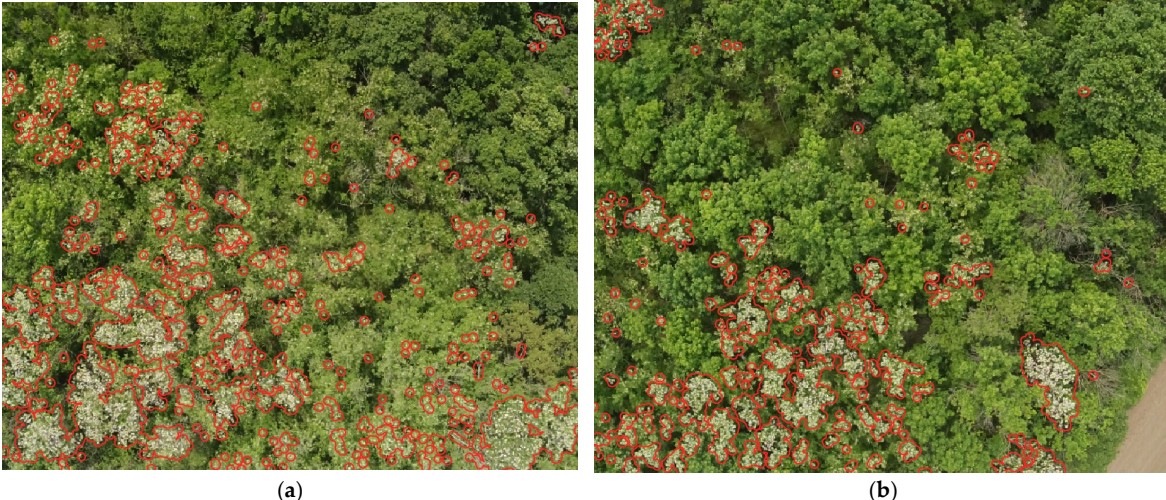

(**a**)                  (**b**)

**Figure 10.** Results from the *EBI* classification and clustering with the proposed algorithm for the first (**a**) and second (**b**) testing images.

**Table 4.** Confusion matrix for the classification of the first testing image using the *EBI* approach.

| | | Reference | | |
|---|---|---|---|---|
| | | Not a White Bloom | White Bloom | Precision |
| Result | Not a white bloom | 7821 | 255 | 0.968 |
| | White bloom | 220 | 1704 | 0.886 |
| | Recall | 0.973 | 0.870 | **Kappa: 0.85** |

**Table 5.** Confusion matrix for the classification of the second testing image the *EBI* approach.

| | | Reference | | |
|---|---|---|---|---|
| | | Not a White Bloom | White Bloom | Precision |
| Result | Not a white bloom | 8301 | 399 | 0.954 |
| | White bloom | 11 | 1289 | 0.992 |
| | Recall | 0.999 | 0.764 | **Kappa: 0.84** |

It can be seen that for the first image, the Kappa is slightly lower (0.85), compared to the one obtained with the developed algorithm (0.88). On the other hand, for the second image, the *EBI*-based Kappa is significantly lower (0.84), compared to the one from the developed algorithm (0.95). Furthermore, the relatively low recall for the "white bloom" class in both images (0.870 and 0.764) indicates that the *EBI* approach returns a number of false positives when classifying non-blooming areas.

The abovementioned results can be explained if we zoom in and look into the details. For example, Figure 11 shows fragments from the first image and the overlapped clusters based on the suggested (red) and the *EBI* (teal) criteria. It can be seen that the *EBI* approach returns additional clusters. Some of them represent small single blooming areas on the trees, as shown in Figure 11a (C, D) and Figure 11b (E, F, G). In most cases, their identification is unwanted because these trees are either not yet entirely blooming or their blooming is already passing. Other clusters wrongly identify logs and branches of withered trees, because of their bright color, i.e., they can be categorized as false-positives. Such examples can be noticed in Figure 11a (A, B, E) and Figure 11b (A, B, C, D).

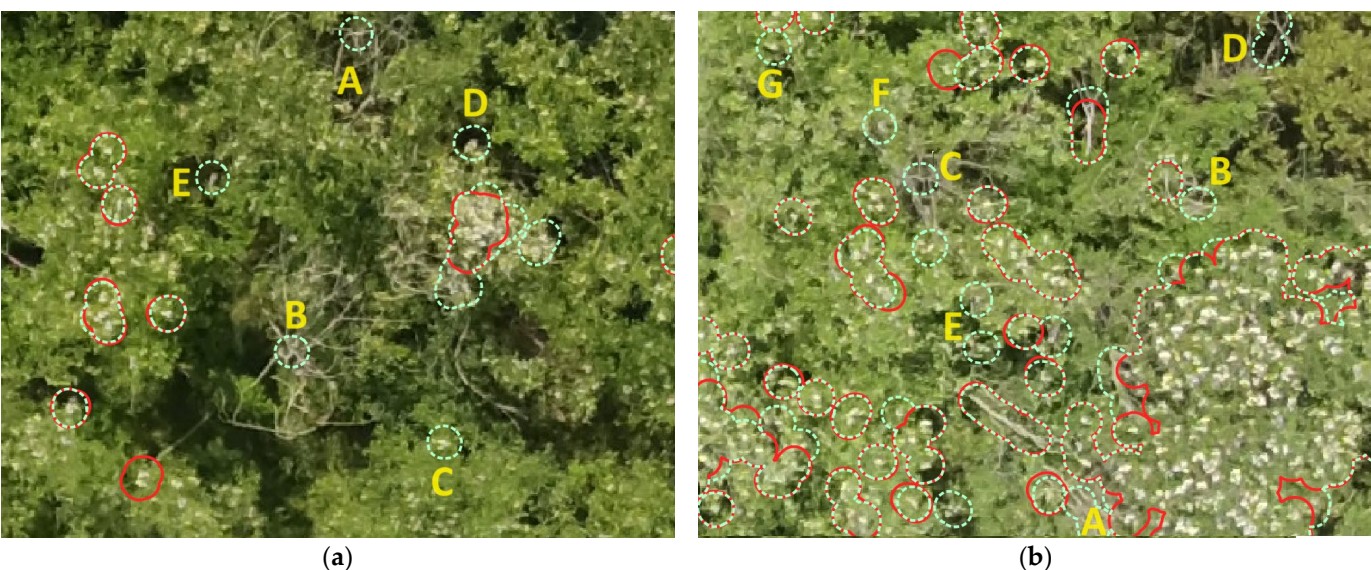

(**a**)  (**b**)

**Figure 11.** Fragments from image 1 with overlapping clusters, identified using the suggested (red) and the *EBI* (teal) criteria: first fragment (**a**); second fragment (**b**).

Close-up fragments from the second testing image are presented in Figure 12. In this case it can be noticed that some of the clusters created by the *EBI* approach do not cover the

entire blooming area and divide it into separate clusters (false negatives). Numerous such examples are presented in Figure 12a (A, B, C, D, E, F) and Figure 12b (A, B, C, D). These details explain the significant difference in the estimated Kappa for the two algorithms when analyzing the second image.

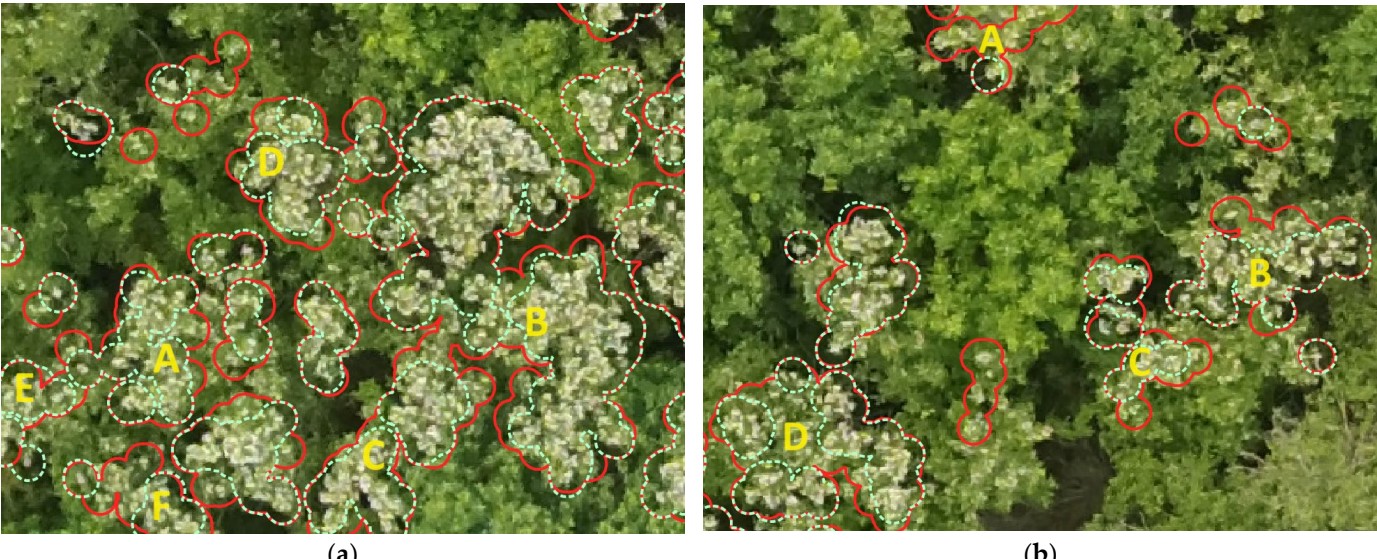

(**a**)              (**b**)

**Figure 12.** Fragments from image 2 with overlapping clusters, identified using the suggested (red) and the *EBI* (teal) criteria: first fragment (**a**); second fragment (**b**).

From the above examples, the following observations can be made:

- If the threshold value for the *EBI* criteria is increased above 1.5, this could potentially reduce the difference between the area of the clusters, estimated with the two criteria for the second image; however, it will also increase the percentage of false positives from the first image, as the area of the clusters, marking the withered trees will also increase.
- If the threshold value for the *EBI* criteria is reduced below 1.5, this could potentially reduce the number of false positives (the identified withered trees) from the first image; however, this will also further reduce the area of the true positive clusters, identified in the second image.

Therefore, it is expected that *EBI* could return a significant share of false negatives and false positives if the observed area has many trunks, branches of withered trees, and partially blooming trees. This allows for the conclusion that the suggested criterion has better performance, compared to the Enhanced Blooming Index, when identifying blooming black locust areas in wild forests.

2. *Comparison with SVM classification*

The supervised pixel-based SVM classification has been chosen as the second comparison method, as it has shown good results with blooming trees in [33]. To implement the classification, we have used the ArcGIS Pro v.3.0.1 software by Esri Inc. (Redlands, CA, USA). The images were first classified into five classes: blooming, trees, trunks, black areas, and soil; and thereafter the last four categories were merged into a single non-blooming class. The results from the classification for the two images are presented in Figure 13 and the corresponding confusion matrices—in Tables 6 and 7. It can be seen that the SVM algorithm returns an enormous number of false positives, which is shown by the relatively low "not a white bloom" recall for both images. They are the reason for the low precision of the "white bloom" class. Furthermore, it can be seen that the Kappa of the SVM-based approach is significantly lower, compared to the other two investigated algorithms, which

indicates that pixel-based SVM classifications are inappropriate for the identification of blooming black locust areas.

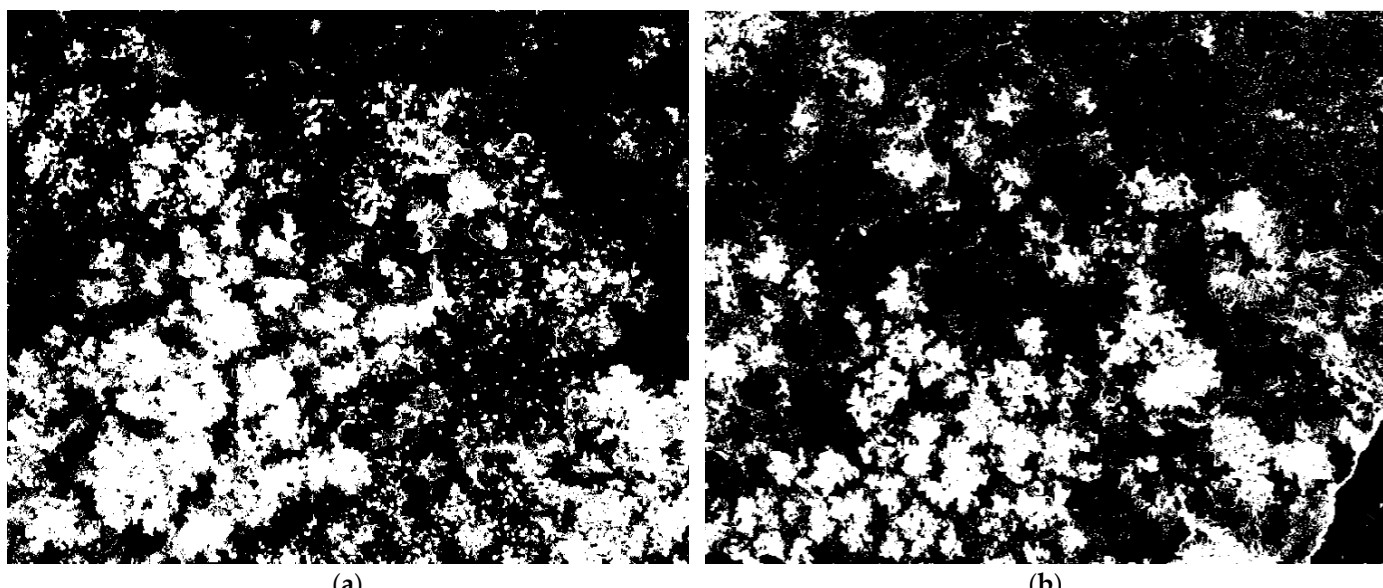

(**a**)                                                              (**b**)

**Figure 13.** Results from the SVM classification for the first (**a**) and second (**b**) images, where white indicates blooming and black—non-blooming areas.

**Table 6.** Confusion matrix for the pixel-based SVM classification of the first testing image.

|        |                   | Reference | | |
|--------|-------------------|-------------------|-------------|------------|
|        |                   | Not a White Bloom | White Bloom | Precision  |
| Result | Not a white bloom | 6937              | 171         | 0.976      |
|        | White bloom       | 1068              | 1788        | 0.626      |
|        | Recall            | 0.867             | 0.913       | **Kappa: 0.66** |

**Table 7.** Confusion matrix for the pixel-based SVM classification of the second testing image.

|        |                   | Reference | | |
|--------|-------------------|-------------------|-------------|------------|
|        |                   | Not a White Bloom | White Bloom | Precision  |
| Result | Not a white bloom | 7173              | 112         | 0.985      |
|        | White bloom       | 1139              | 1576        | 0.581      |
|        | Recall            | 0.863             | 0.934       | **Kappa: 0.64** |

*3.3. Limitations*

The developed criterion (Equation (3)) is only appropriate for the identification of white blooming trees, such as black locust (*Robinia pseudo-acacia*), hawthorn (*Crataegus monogyna)*, wild plum (*Prunus cerasifera)*, rosehip (*Rosa canina)*, wild pear (*Pyrus pyraster*), etc. It does not perform well with other colors of vegetation, such as yellow (Sunflower), red (Poppy), etc. Furthermore, the developed approach might create false positive results if the observed areas contain bright artificial (such as a cement walkway/path) or natural (such as water, stones, parts of the sky, etc.) objects.

It should be considered that the influence of the weather and lighting conditions has not been investigated in this study. Therefore, it might be necessary to perform additional preprocessing of the images, such as adjusting their brightness, in order to obtain accurate results. Furthermore, in this study, it is accepted that there are no partial shadows over the investigated area, created by clouds. If such shadows exist, additional preprocessing might be also required.

*3.4. Scientific and Practical Contributions of the Study*

This study contributes both scientifically and practically. Its scientific contribution includes the introduction of an efficient algorithm for white blooming tree identification, systematically comparing it with existing methods, and highlighting its advantages in terms of identifying blooming black locust trees in wild forests. The algorithm showcases the ability to identify blooming trees with a nuanced approach, acknowledging areas containing soil or withered trees and partially blooming black locust trees, contributing to a more detailed and accurate recognition process. Furthermore, it allows for the estimation of the percentile area of each class, providing a quantitative measure of the distribution of white blooming trees in the studied area.

The practical value of the study is that instructions are given for the implementation of the developed algorithm in the open-source QGIS v.3 software, making it accessible and applicable in real-world scenarios. Furthermore, the study provides practical insights into the limitations of the developed criterion, specifying its applicability to certain types of flowering trees and potential challenges related to environmental conditions.

## 4. Conclusions

The current study presents a new algorithm for the identification and delineation of white flowering honey plants in mixed forest ecosystems using UAV-based RGB imaging. To verify the reliability of recognition, mixed forest massifs of white flowering honey plants, such as black locust (*Robinia pseudo-acacia*) and other plant species, which are not suitable for beekeeping, were selected in the area of the village of Yuper, North-eastern Bulgaria. The selected arrays were scanned using a five-band multispectral DJI Phantom drone flying at a low altitude of 64 m, which provided high-resolution images.

The developed detection algorithm has been implemented in the software QGIS v.3. The suggested criterion was found to effectively distinguish white from non-white colors. Its overall accuracy was assessed using Cohen's Kappa, whose values were 0.88 and 0.95, respectively, for the two testing images. This allows us to conclude that the developed algorithm can detect white flowering honey trees in mixed forest ecosystems with very high accuracy. Furthermore, the algorithm meets all the defined requirements. The classification results showed that approximately 14–15% of the investigated forest represents white-flowered trees, and the remaining 86–85% is something else.

The efficiency of the algorithm was compared with two other approaches: the first one uses the Enhanced Bloom Index, and the second one is a supervised pixel-based SVM classification. It was found that the *EBI* approach generates false positives and false negatives when the investigated area has partially blooming trees, trunks, branches of withered trees, and other bright objects (natural or artificial), which could potentially lead to significant errors. The abovementioned inaccuracies are avoided or at least limited by the novel algorithm developed in our research. The confusion matrices of all algorithms were obtained using 10,000 randomly chosen pixels within the previously selected ROIs. The Kappa for the *EBI*-based approach ranges from 0.84 to 0.85 and for the SVM classification—from 0.64 to 0.66. These results clearly show that the developed algorithm detects blooming black locusts in wild forests with higher accuracy, compared to the reference methods. Furthermore, considering the low Kappa for the SVM classification we can conclude that supervised pixel-based classifications are not appropriate for the identification of white blooming trees in wild forests.

Our future research on the investigated topic will be focused in several directions: (1) Developing automated and real-time methods for more efficient and accurate recognition of white blooming trees; (2) Testing the developed criterion and algorithm with other white blooming trees (*Crataegus monogyna*, *Prunus cerasifera*, *Rosa canina*, *Pyrus pyraster*), with satellite-obtained images, and under different environmental conditions; (3) Developing new algorithms, which are able to recognize differently colored honey-flowering trees in mixed forest ecosystems.

**Author Contributions:** Conceptualization, A.Z.A. and B.I.E.; methodology, A.Z.A. and B.I.E.; software, B.I.E.; validation, A.Z.A., B.I.E. and V.N.V.; formal analysis, B.I.E.; investigation, A.Z.A.; resources, A.Z.A.; data curation, B.I.E.; writing—original draft preparation, A.Z.A. and B.I.E.; writing—review and editing, B.I.E., V.N.V. and S.-S.B.; visualization, B.I.E.; supervision, V.N.V. and S.-S.B. All authors have read and agreed to the published version of the manuscript.

**Funding:** The APC was funded by the National University of Science and Technology POLITEHNICA Bucharest, Splaiul Independentei, Nr 313, Sector 6, 006042 Bucharest.

**Institutional Review Board Statement:** Not applicable.

**Informed Consent Statement:** Not applicable.

**Data Availability Statement:** The data presented in this study are available on request from the corresponding author.

**Acknowledgments:** This research is supported by the Bulgarian National Science Fund under Project KP-06-PN 46-7 "Design and research of fundamental technologies and methods for precision apiculture".

**Conflicts of Interest:** The authors declare no conflicts of interest.

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
