# Peer review of "A Novel Algorithm to Detect White Flowering Honey Trees in Mixed Forest Ecosystems Using UAV-Based RGB Imaging"

_agriengineering, doi:10.3390/agriengineering6010007_

Round 1

Reviewer 1 Report

Comments and Suggestions for Authors

This paper outlines a study focused on utilizing drone-based RGB imaging technology to develop an algorithm for identifying and distinguishing white nectar source plants, particularly black locust, in mixed forest ecosystems. The research employed a DJI Phantom 4 multispectral drone equipped with six cameras, monitoring the plant coverage of black locusts during the May 2023 growing season in the village of Jupiter, northeastern Bulgaria. The geographical location, drone-captured RGB and multispectral images, and subsequent analysis using QGIS 3.0 software are detailed. The newly created standard for distinguishing white and non-white plants was applied, revealing that approximately 14%-15% of areas were classified as white flower trees. The algorithm's performance was compared to other methods, demonstrating its user-friendly nature for individuals with limited technical experience. The proposed method holds significant importance for beekeepers in assessing the production potential and selecting suitable beekeeping sites in mixed forest ecosystems.

The paper is logical and the experimental setup is sound, but there are still some problems. Deficiencies and possible directions for improvement of this article include:

1.Automation and real-time: the method in the article relies heavily on manually setting thresholds, which can lead to human-influenced recognition results. Future research could explore automation and real-time methods for more efficient and accurate recognition.

2.Environmental factors: The article has not yet investigated the effects of weather and lighting conditions on recognition results. Future research could consider image preprocessing, such as adjusting brightness, for more accurate results.

3.Scope of application: The article has not yet tested the performance of the method on other white flowering trees, satellite-acquired images, and under different environmental conditions. Future research can further extend the application scope to verify the effectiveness of the method in a wider range of scenarios.

4.Algorithm optimization: Although the article presents an effective recognition method, there is still room for optimization. Future research can explore more efficient and accurate algorithms to improve the performance of recognizing white flowers and trees.

5.Field validation: The recognition results in the article mainly rely on the images acquired by the UAV. Future research could conduct field validation to ensure the accuracy and reliability of the recognition results.

6.Integration with other techniques: future research could explore the integration of the method with other remote sensing techniques (e.g., LiDAR) to obtain more comprehensive and accurate information on the distribution of nectar plants.

Overall, the article could be improved by addressing these shortcomings and providing more comprehensive validation and comparison with existing methods.

Comments on the Quality of English Language

This paper outlines a study focused on utilizing drone-based RGB imaging technology to develop an algorithm for identifying and distinguishing white nectar source plants, particularly black locust, in mixed forest ecosystems. The research employed a DJI Phantom 4 multispectral drone equipped with six cameras, monitoring the plant coverage of black locusts during the May 2023 growing season in the village of Jupiter, northeastern Bulgaria. The geographical location, drone-captured RGB and multispectral images, and subsequent analysis using QGIS 3.0 software are detailed. The newly created standard for distinguishing white and non-white plants was applied, revealing that approximately 14%-15% of areas were classified as white flower trees. The algorithm's performance was compared to other methods, demonstrating its user-friendly nature for individuals with limited technical experience. The proposed method holds significant importance for beekeepers in assessing the production potential and selecting suitable beekeeping sites in mixed forest ecosystems.

The paper is logical and the experimental setup is sound, but there are still some problems. Deficiencies and possible directions for improvement of this article include:

1.Automation and real-time: the method in the article relies heavily on manually setting thresholds, which can lead to human-influenced recognition results. Future research could explore automation and real-time methods for more efficient and accurate recognition.

2.Environmental factors: The article has not yet investigated the effects of weather and lighting conditions on recognition results. Future research could consider image preprocessing, such as adjusting brightness, for more accurate results.

3.Scope of application: The article has not yet tested the performance of the method on other white flowering trees, satellite-acquired images, and under different environmental conditions. Future research can further extend the application scope to verify the effectiveness of the method in a wider range of scenarios.

4.Algorithm optimization: Although the article presents an effective recognition method, there is still room for optimization. Future research can explore more efficient and accurate algorithms to improve the performance of recognizing white flowers and trees.

5.Field validation: The recognition results in the article mainly rely on the images acquired by the UAV. Future research could conduct field validation to ensure the accuracy and reliability of the recognition results.

6.Integration with other techniques: future research could explore the integration of the method with other remote sensing techniques (e.g., LiDAR) to obtain more comprehensive and accurate information on the distribution of nectar plants.

Overall, the article could be improved by addressing these shortcomings and providing more comprehensive validation and comparison with existing methods.

Author Response

Response to Reviewer 1 Comments

This paper outlines a study focused on utilizing drone-based RGB imaging technology to develop an algorithm for identifying and distinguishing white nectar source plants, particularly black locust, in mixed forest ecosystems. The research employed a DJI Phantom 4 multispectral drone equipped with six cameras, monitoring the plant coverage of black locusts during the May 2023 growing season in the village of Jupiter, northeastern Bulgaria. The geographical location, drone-captured RGB and multispectral images, and subsequent analysis using QGIS 3.0 software are detailed. The newly created standard for distinguishing white and non-white plants was applied, revealing that approximately 14%-15% of areas were classified as white flower trees. The algorithm's performance was compared to other methods, demonstrating its user-friendly nature for individuals with limited technical experience. The proposed method holds significant importance for beekeepers in assessing the production potential and selecting suitable beekeeping sites in mixed forest ecosystems.

The paper is logical and the experimental setup is sound, but there are still some problems. Deficiencies and possible directions for improvement of this article include:

Point 1: Automation and real-time: the method in the article relies heavily on manually setting thresholds, which can lead to human-influenced recognition results. Future research could explore automation and real-time methods for more efficient and accurate recognition.

Response 1: Thank you for your thoughtful review. We appreciate your insight regarding the reliance on manually set thresholds in our method. We acknowledge the potential influence of human factors on recognition results and agree that exploring automation and real-time methods could enhance efficiency and accuracy in future research. We will carefully consider your suggestion and aim to incorporate these aspects into our future work to further improve the robustness of our recognition approach. Your feedback is invaluable, and we look forward to addressing these concerns in our future research endeavors. Added in manuscript conclusion

Point 2:.Environmental factors: The article has not yet investigated the effects of weather and lighting conditions on recognition results. Future research could consider image preprocessing, such as adjusting brightness, for more accurate results.

Response 2: We appreciate your observation regarding the environmental factors, specifically the effects of weather and lighting conditions on our recognition results. Your suggestion to explore image preprocessing, including brightness adjustment, is duly noted.

In our experiment, flight operations were carried out from 12:00 a.m. to 13:00 p.m. to minimize shading while flying when lighting conditions were best according to our portable agrometeorological station at these hours the average solar radiation was 513 W m-2. During the experiment the meteorological conditions remained steady and there was no viewable cloud clover.

In our future research, we plan to delve into the impact of these environmental variables and implement suitable image preprocessing techniques to enhance the accuracy and robustness of our recognition system. But this would also require specially conducted experiments under different meteorological conditions. Added in manuscript conclusion.

Point 3: Scope of application: The article has not yet tested the performance of the method on other white flowering trees, satellite-acquired images, and under different environmental conditions. Future research can further extend the application scope to verify the effectiveness of the method in a wider range of scenarios.

Response 3: Your suggestion to extend the application scope is well-taken, and we recognize the importance of verifying the effectiveness of our method in a wider range of scenarios.

Unfortunately, during the evaluation of the white flowering plant recognition algorithm, there was only black locust (Robinia pseudo-acacia), because due to the seasonality of flowering, the other white flowering species, such as hawthorn (Crataegus monogyna), wild plum (Prunus cerasifera), rosehip (Rosa canina), wild pear (Pyrus pyraster) had overbloomed earlier. For the next season, in our future research endeavors, we plan to address these considerations by conducting comprehensive testing on various white flowering trees, analyzing satellite-acquired images, and exploring performance under different environmental conditions. Added in manuscript conclusion

Point 4: Algorithm optimization: Although the article presents an effective recognition method, there is still room for optimization. Future research can explore more efficient and accurate algorithms to improve the performance of recognizing white flowers and trees.

Response 4: Thank you for your thoughtful review and valuable feedback. In our future research, we plan to optimize our algorithm by validating it under different environmental conditions and with different white-flowering species. We are committed to addressing this optimization aspect to ensure our method reaches its full potential. Added in manuscript conclusion

Point 5: Field validation: The recognition results in the article mainly rely on the images acquired by the UAV. Future research could conduct field validation to ensure the accuracy and reliability of the recognition results.

Response 5: We appreciate your emphasis on the importance of field validation to ensure the accuracy and reliability of our recognition results. The specificity of our research in mixed forest ecosystems suggests the use of UAV-based RGB imaging because unmanned aerial vehicles (UAVs) equipped with high-resolution sensors able to detect small objects, such as flowering plants, because of the ability to collect frequently very precise high-resolution spatial images. Using stationary cameras in the field to validate the accuracy of the results is difficult due to the inability to spatially cover the white flowering plants in the mixed forest. We plan to validate the algorithm on site with meadow flowering vegetation, where it is possible to photograph it with a stationary camera.

Point 6: Integration with other techniques: future research could explore the integration of the method with other remote sensing techniques (e.g., LiDAR) to obtain more comprehensive and accurate information on the distribution of nectar plants.

Response 6: Appreciating the recommendations made highly, we have added them to the manuscript in conclusions as a target for our future research.

Overall, the article could be improved by addressing these shortcomings and providing more comprehensive validation and comparison with existing methods.

Response 7: In order to address the validation issue, in the new version of the article we have explained the validation methodology and have added the resulting confusion matrices for each classification method and testing object. This way we can compare the different methods not only graphically but also statistically.

Reviewer 2 Report

Comments and Suggestions for Authors

„A novel algorithm to detect white flowering honey trees in 2 mixed forest ecosystems using UAV-based RGB imaging” was investigated in this study.

Main questions was aimed of this study (which in this paper, the authors evidently consider an innovation):

(1) to create a novel algorithm to identify and distinguish white flowering honey plants, such as black locust (Robinia pseudo-acacia) and,

(2) to determine the areas occupied by this forest species in mixed forest ecosystems using UAV-based RGB imaging.

The main questions and objectives of this research are very well described and defined in the submitted manuscript.

It would be necessary for the authors to highlight more clearly what is new:

- new scientific contributions and,

- new practical contributions,

in their research.

It is not clearly pointed out what is new scientific and/or practical contributions in their study.

The topic in this study is original, relevant and specific in this field of research.

However, the authors should highlight this in more detail, in the manuscript itself: originality, relevance, specificity, novelty..., of their research topics in this area.

I note once again, there is a detailed description, but you just need to clearly highlight the key points.

The manuscript is interesting and deals with a current and innovative topic in the computer vision for agriculture and smart farming, and etc. topics.
However, i would suggest the authors to improve the manuscript in the Introduction so that it contains more data on similar research, which would further emphasize the importance of the topic of this manuscript in relation to similar previously published materials.

The methodology in this manuscript is very interesting and adequately described and presented.

The method of data collection chosen in experimental work can always be a subject of discussion.

In this case, I believe that the collection methods of data, which the authors selected and described, was adequate.

This is very important in the case of such and similar researches.

This is also significant for the potential readers of this paper, such as "AgriEngineering".

The results are adequately presented in chapter 3. Results.

A discussion of the results is good and is given in chapter 4. Discussion.

Тhere is a chapter on conclusions, 5. Conclusions, and it is included after extensive discussion. The present conclusion is comprehensive and adequately designed and it's not too long. But what is said is relevant.

However, it could be improved when all the improvements according to the previous points are followed in order.

The listed references are appropriate. Considering the above, it may be necessary for the authors to improve and supplement them.

Although I am not an expert in English, I think the English in this manuscript can be improved.

Considering the above, this is a request for „Ðœajor revisions”, looking at the comments I made in the first part of my review.

Author Response

Response to Reviewer 2 Comments

„A novel algorithm to detect white flowering honey trees in 2 mixed forest ecosystems using UAV-based RGB imaging” was investigated in this study.

Main questions was aimed of this study (which in this paper, the authors evidently consider an innovation):

(1) to create a novel algorithm to identify and distinguish white flowering honey plants, such as black locust (Robinia pseudo-acacia) and,

(2) to determine the areas occupied by this forest species in mixed forest ecosystems using UAV-based RGB imaging.

The main questions and objectives of this research are very well described and defined in the submitted manuscript.

It would be necessary for the authors to highlight more clearly what is new:

- new scientific contributions and,

- new practical contributions,

in their research.

It is not clearly pointed out what is new scientific and/or practical contributions in their study.

Response: Thank you for your thoughtful review and valuable feedback. In order to address your recommendations, we have pointed out the novelty and the achieved at different places in the discussion and the conclusions. Furthermore, we have created the section “3.4. Scientific and practical contributions of the study”, which summarizes the scientifical and practical novelty of the study.

The topic in this study is original, relevant and specific in this field of research.

However, the authors should highlight this in more detail, in the manuscript itself: originality, relevance, specificity, novelty..., of their research topics in this area.

I note once again, there is a detailed description, but you just need to clearly highlight the key points.

The manuscript is interesting and deals with a current and innovative topic in the computer vision for agriculture and smart farming, and etc. topics.
However, i would suggest the authors to improve the manuscript in the Introduction so that it contains more data on similar research, which would further emphasize the importance of the topic of this manuscript in relation to similar previously published materials

The methodology in this manuscript is very interesting and adequately described and presented.

The method of data collection chosen in experimental work can always be a subject of discussion.

In this case, I believe that the collection methods of data, which the authors selected and described, was adequate.

This is very important in the case of such and similar researches.

This is also significant for the potential readers of this paper, such as "AgriEngineering".

The results are adequately presented in chapter 3. Results.

A discussion of the results is good and is given in chapter 4. Discussion.

Тhere is a chapter on conclusions, 5. Conclusions, and it is included after extensive discussion. The present conclusion is comprehensive and adequately designed and it's not too long. But what is said is relevant.

However, it could be improved when all the improvements according to the previous points are followed in order.

Response: We believe we have improved the manuscript and the objectives achieved have been given in the conclusions. We have added a validation of the methods, which is explained in section “3.1. Data processing”. For each of the developed models there is now a confusion matrix and the accuracy of the algorithms is evaluated using Cohen’s Kappa.

The listed references are appropriate. Considering the above, it may be necessary for the authors to improve and supplement them.

Although I am not an expert in English, I think the English in this manuscript can be improved.

Considering the above, this is a request for „Ðœajor revisions”, looking at the comments I made in the first part of my review.

Reviewer 3 Report

Comments and Suggestions for Authors

The only chapter on which I have no significant comments is Chapter 1. Introduction.

Chapter. 3. Results and Discussion, contains too much information on the used methodology, so this information should be moved to Chapter 2. Materials and Methods. Furthermore, the results are presented in Chapter 3 in an unprofessional manner. It is not stated how many images were tested in the comparisons. There is only information that two images were tested for each method, but it is unclear if it was always the same images for all methods. In the tables, there are percentage values without any indication of the measurement error resulting from the results of the corresponding number of images analysed. In such a situation, the results have no scientific value. When testing image classification methods, one should compare several sample images and then evaluate the effectiveness of both methods statistically. 

Furthermore, the discussion is not only about comparing the results of one's research but also about referring to results or methods found in other published research papers, especially those referred to in Chapter 1. Introduction. There are too few of these references in the chapter. This may be due to insufficiently inquisitive analysis of the results in context to existing, already published studies.

The chapter Conclusions should be improved. Its content should contain the results' conclusions and possible suggestions for further research. In this chapter, the content mainly consists of a summary of the article. The content in this chapter should refer to the aim of the research, which is not here. Moreover, the research aimed to create a novel algorithm to identify and distinguish white flowering honey plants, such as black locust (Robinia pseudo-acacia), and to determine the areas occupied by this forest species in mixed forest ecosystems using UAV-based RGB imaging.

The authors should have commented on whether they achieved their objective, as the manuscript only described that they compared several methods.

The authors admitted that they did not consider weather and lighting conditions but did not confirm they were always the same when taking the images.

Furthermore,

Abbreviations used in the manuscript should be preceded by the full name when first used. 

The authors should consider improving the manuscript to give a concrete answer in the conclusions regarding the objective achieved.

Author Response

Response to Reviewer 3 Comments

The only chapter on which I have no significant comments is Chapter 1. Introduction.

Point 1: Chapter. 3. Results and Discussion, contains too much information on the used methodology, so this information should be moved to Chapter 2. Materials and Methods.

Response 1: The first part of section 3.1, which explains the implementation methodology, has been moved to section 2.3.

Point 2: Furthermore, the results are presented in Chapter 3 in an unprofessional manner. It is not stated how many images were tested in the comparisons. There is only information that two images were tested for each method, but it is unclear if it was always the same images for all methods. In the tables, there are percentage values without any indication of the measurement error resulting from the results of the corresponding number of images analysed. In such a situation, the results have no scientific value. When testing image classification methods, one should compare several sample images and then evaluate the effectiveness of both methods statistically. 

Response 2: Thank you for the valuable comment. We have improved Chapter 3 according to the made recommendations with the following key changes:

  1. We have tried to explain better the procedure for selection of the testing images in section 3.1. Furthermore, in section 3.2 we have explicitly stated that “in order to be able to directly compare the accuracy of the developed algorithm with the other two approaches, the same testing images were used”;
  2. We have added confusion matrices for all classification algorithms and have explained the procedure for validation of the models. The classification results are now analyzed according to the obtained Precision, Recall and Cohen’s Kappa.

Point 3: Furthermore, the discussion is not only about comparing the results of one's research but also about referring to results or methods found in other published research papers, especially those referred to in Chapter 1. Introduction. There are too few of these references in the chapter. This may be due to insufficiently inquisitive analysis of the results in context to existing, already published studies.

Response 3: We appreciate your observation regarding the other published research papers. To the best of our knowledge the only study which have suggested an approach for identifying white blooming trees is reference [33]. That is why we used the suggested Enhanced Bloom Index as one of the reference methods. The second reference method is a generic one – supervised SVM classification. In the introduction are cited many studies, which deal with object-based identification of blooming trees, which means they are actually identifying the blooming flower as an object or a separate tree in an orchard. This approach is not applicable in our case, because the Black Locust flowers are too small and too many. This is described in section 2.3, when explaining the approach, we have chosen. For the abovementioned reasons, there is only one study, which we can use as a reference one and we do use it.

Point 4: The chapter Conclusions should be improved. Its content should contain the results' conclusions and possible suggestions for further research. In this chapter, the content mainly consists of a summary of the article. The content in this chapter should refer to the aim of the research, which is not here. Moreover, the research aimed to create a novel algorithm to identify and distinguish white flowering honey plants, such as black locust (Robinia pseudo-acacia), and to determine the areas occupied by this forest species in mixed forest ecosystems using UAV-based RGB imaging.

The authors should have commented on whether they achieved their objective, as the manuscript only described that they compared several methods.

Response 4: We have reorganized the manuscript and rewritten the conclusions section in order to better summarize the obtained results. We have also clearly stated that the aims of the research have been met.

Point 5: The authors admitted that they did not consider weather and lighting conditions but did not confirm they were always the same when taking the images.

Response 5: Thank you for pointing out that this information was missing. In section 2.2 are described the environmental conditions during the UAV experiments, and the lighting conditions were the same for the hours when the images were taken.

Point 6: Furthermore,

Abbreviations used in the manuscript should be preceded by the full name when first used. 

Response 6: Some missing definitions of abbreviations have been added.

Point 7: The authors should consider improving the manuscript to give a concrete answer in the conclusions regarding the objective achieved.

Response 7: We believe we have improved the manuscript and the objectives achieved have been given in the conclusions.

Round 2

Reviewer 2 Report

Comments and Suggestions for Authors

The reviewers' comments are noted.

The manuscript has now been significantly improved.